# Is the Relationship between Acute and Chronic Workload a Valid Predictive Injury Tool? A Bayesian Analysis

**DOI:** 10.3390/jcm11195945

**Published:** 2022-10-08

**Authors:** Leandro Carbone, Matias Sampietro, Agustin Cicognini, Manuel García-Sillero, Salvador Vargas-Molina

**Affiliations:** 1Department of Physical Education and Sport, Faculty of Medicine, University of Salvador, Buenos Aires C1020ADN, Argentina; 2Physiotheraphy Department Belgrano Football Club, Nacional University of Cordoba, Cordoba X5000HUA, Argentina; 3Department of Physical Education and Sport, Faculty of Sport Sciences, EADE-University of Wales Trinity Saint David, 29018 Malaga, Spain; 4Physical Education and Sport, Faculty of Medicine, University of Málaga, 29010 Malaga, Spain

**Keywords:** injuries, ACWR, perceived exertion, workload, performance, prevention strategies

## Abstract

This study aimed to evaluate the relationship between injury risk, acute load (AL), acute chronic workload ratio (ACWR) and a new proposed ACWR. Design: a retrospective cohort study of the year 2018 was conducted on Argentine first-division soccer players. Participants: Data from 35 players (age = 26.7 ± 4.71 years; height = 176.28 ± 6.09 cm; mass = 74.2 ± 5.27 kg) were recorded; 12 players’ data were analyzed for 1 year, and 23 players’ data were analyzed for 6 months. Interventions: The mean difference of ACWR (MD = 0.22), high-density interval (HDI 95% = (0.07, 0.36)) and AL (MD = 449.23, HDI 95% = (146.41, 751.2)) between groups turned out to be statistically significant. The effect size between groups comparing ACWR and AL was identical (ES = 0.64). Results: The probability of suffering an injury conditioned by ACWR or random ACWR was similar for all estimated quantiles, and the differences between them were not statistically significant. Conclusions: The ACWR ratio, using internal load monitoring, is no better than a synthetic ACWR created from a random denominator to predict the probability of injury. ACWR should not be used in isolation to analyze the causality between load and injury.

## 1. Introduction

The injury rate in today’s modern soccer has generated constant concern, both in the scientific field as well as in the training staff in elite teams, due to the negative impact higher injury rates may have on success, given that a negative correlation has been found between injury rates and the league ranking of European professional teams [1]. Injuries involve a high economic cost resulting from the loss of participation of a professional player; this cost has been estimated at €500,000 per month [2]. On the other hand, player availability is positively associated with team success; therefore, injury prevention is essential in professional soccer clubs and international sports federations, resulting in a crucial task for medical and coaching teams [1,3]. For this reason, multidimensional approaches based on injury management and preventive strategies are prioritized [4].

Modern soccer is played at a higher intensity, with faster game criteria, and may even be played more aggressively from a physical point of view than in previous decades [5]. Additionally, the physical demands of the game involve an average of 11 km of total distance covered per match, distributed over a wide range of speed thresholds, such as maximum intensity short sprints (<10 m), accounting for approximately 12% of the total distance covered within a match, and very high-intensity accelerations and decelerations, approximately 25 accelerations (>4 m/s^2^) and 45 decelerations (>−4 m/s^2^) [6,7,8]. These data represent an increased demand on players’ musculoskeletal structures and connective tissue to optimize performance and lower injury risk. As a result of this increase in physical demands in competition, the phenomenon of injury incidence in elite soccer players has increased in recent years [9].

Nevertheless, it is essential to know the relationships between internal load (IL) and external load (EL) to understand the dose-response relationship between training and competition. IL measures derived from session perceived exertion (sRPE) have shown a positive association with external loads derived from meters run at different speeds and acceleration thresholds. It has been reported that the total running distance has the strongest association with sRPE monitoring values [10]. Therefore, it is possible to validly and reliably use sRPE to quantify training intensity in a dose-response relationship in soccer [11,12]. The relationship between TL, performance and injury risk has been previously documented [1,13]. Moreover, not only the rating of perceived exertion but also other psychological factors, such as beliefs, previous experiences and attitudes, may have a negative impact on the injury risk of high-level soccer players [14].

Since Borg [15] proposed the usefulness of assessing perceived exertion for quantifying training load, different scales have proven to be useful for estimating intra-session workloads in both strength training [16] and cardio-respiratory training [17,18]. In addition, RPE has been used to self-adjust loads from set to set, in combination with repetitions in reserve during resistance training [19]. Moreover, Foster et al. introduced the concept of session rating of perceived exertion (session-RPE), whereas the intensity and duration of the session were assessed [20]. In sports science literature, the distinction between internal and external load is commonly made. The external load refers to the load the players have actually been exposed to (total distance, sprinting distance, etc.), whereas the internal load refers to how the players perceive this load (heart rate, RPE, etc.) [12]. By seeing the external and internal variables in relation to each other, we can get insights into how a given external stressor is assimilated by an athlete at a given time. While time was reported in minutes, the intensity was monitored by self-answer to the question “How was your training?” using a scale from 1 to 10, albeit modified and referring to the perception of the average intensity of the session. In this way, the arbitrary units (AU) would reflect the TL of the training session (e.g., 60 min × RPE 4 = 240 AU). Additionally, from the information obtained in AU at the end of a series of consecutive sessions, another value can be obtained, such as monotony (MON), which refers to the variation of training day after day; in fact, the occurrence of overtraining is related to MON and the application of high TLs [21], For this calculation, MON would be the weekly mean TL divided by the standard deviation (SD) (monotony = weekly mean TL/SD). Therefore, it can be additional information to take into account from an injury prevention standpoint.

The relationship between sports injury and TL was hypothesized by Kibler et al. more than 30 years ago [22]. There is extensive literature on parameters and predictive models on individuals’ risk of injury by monitoring both objective and subjective variables [23]. Recent research on the existing relationship between TL and performance has examined the absolute workload performed in one week (referred to as acute workload) relative to the 4-week chronic workload (i.e., the 4-week average of the acute workload) [24]. The use of the acute: chronic workload ratio (ACWR) to manage changes in load and how these changes relate to injury risk has received increasing attention from the scientific community in recent years [25].

The ACWR is calculated as the acute (i.e., recent) load (AL) divided by the chronic (i.e., long-term) load [25]. Acute to random workload ratio is ‘as’ associated with injury as acute to actual chronic workload ratio: time to dismiss ACWR and its components and was developed as a preventive model to predict injury.

Recently, attempts have been made to make explicit the relationship between the training load to which the athlete is being exposed in relation to the training load for which he/she is adapted [26].

Previous studies on professional soccer and rugby players have shown that high chronic workloads with large peaks in acute workloads (high changes in ACWR) are related to high injury risk [27,28,29,30]. Despite the popularity of ACWR in monitoring TL and its relationship to injury, there is conflicting data regarding its ability to predict and determine causality in sports injuries. Associations have been found between high ACWRs and increased injury risk in elite soccer [31], but no association has been found between increased ACWRs and increased injury risk in sports such as rugby.

We hypothesize that the use of ACWR is not a correct predictor to explain the relationship between load and injury. To check this, we create a new metric ACWRr, which lacks the foundations for which the ACWR is supposed to have predictive power on injuries. If ACWR were a valid tool for predicting injury risk, it should show a strong correlation with observed injuries, and the new metric ACWRr should show less or no relationship with injuries. Hierarchical Bayesian models were used to access the probability distributions associated with the model parameters, thus facilitating the interpretation of the results/relationship.

## 2. Materials and Methods

To examine the relationship between injury risk, acute load, ACWR and a random ACWR created from the replacement of the ACWR, a retrospective cohort study of elite soccer players who were part of one team at the highest level of the Argentine competition during the 2018 year was analyzed. Data were collected from 35 players (average ± SD, age: 26.7 ± 4.71 years; height: 176.28 ± 6.09 cm; mass: 74.2 ± 5.27 kg), of which 12 were followed by 1 year and 23 for 6 months.

All time-loss injuries were recorded. A time-loss injury was defined as any event resulting in a cessation of at least one training session or competition. All injuries that prevented the players from fully participating in all training and competition activities normally planned for that day and prevented participation for a period of more than 24 h were recorded. This classification//definition conforms to the consensual definitions of time-loss injury proposed for the sports athlete team [1,32]. The severity of injuries was recorded according to the time lost, defined as the number of days that the athlete was unavailable for training and competition, from the date of onset until the athlete had resumed training and competition [32]. All injuries were classified by the medical and physiotherapy staff [32,33]. Injuries were also classified by type of injury (according to Orchard Sports Injury Classification System (OSICS) version 10.0), place of body (location of injury), injury mechanism, and mode of onset [32]; these data were collected according to consensus data collection guidelines for information collection from epidemiological studies, including STROBE-SIIS standards. Time loss due to illness or pathophysiologic episodes was not included.

The intensity of all training sessions (including rehabilitation sessions) and match-play was estimated using Borg CR-10’s modified RPE, with scores obtained from each individual player immediately after the end of each training session and match. Each player had the scale explained before the start of the season, and players were asked to report their RPE for each session confidentially without knowledge of other players’ ratings [31,33]. During the 2018 year, players reported their RPE individually every training session and were matched to the same researcher after seeing a printed scale in a face-to-face manner [33]. Session-RPE in arbitrary units (AU) for each player was derived by multiplying RPE and session duration (min). s-RPE has previously been shown to be a valid method for estimating exercise intensity time period [11,20].

### Statistical Analysis

Bayesian hierarchical models were applied to relate load types and injury incidence. The use of these models represents an advantage over the frequentist paradigm, as they allow direct interpretation of their results since they allow access to the posterior distributions of their parameters [34,35]. One of the goals of the present study was to compare the mean of the ACWR and acute load data of the individuals who suffered injuries with those who did not suffer injuries. Usually, in a frequentist framework, we should determine if the data we want to compare follows a normal distribution or not to determine what type of test we should perform. Under the Bayesian framework, if we want to compare the means of the data, we must raise the assumption about the distribution from which the data come to be able to estimate their parameters and compare them. During the exploratory analysis of the data, we noticed that the ACWR and acute load distributions were symmetric with respect to the mean but presented a higher density of data far from the center of the distribution compared to what would be expected from a normal distribution. A Shapiro–Wilk test was performed to assess the normality of both data sets. In both cases, the hypothesis that the data came from a normal distribution was rejected (*p*-value < 0.05). In these cases, using a normal likelihood to estimate parameters could lead to an overestimation of the scale parameter (the standard deviation) and a possible bias in the location parameter (the mean). One of the solutions that can be carried out in these cases is to exchange the use of a normal distribution for another that can handle the outliers in the data. The Student’s t distribution is usually presented as a solution and is widely used to perform robust regressions in a Bayesian setting [36,37,38]. The normal distribution and the Student’s t are two very similar distributions, symmetric about their mean. Both have a location parameter (Mu) that corresponds to the mean of the distribution, and both have a scale parameter (Sigma); the main difference is that the Student’s t distribution has a third parameter, usually called degrees of freedom (Nu), that allows the weight of the tails to be controlled. The larger Nu is, the thinner tails the distribution has, and the more it approaches a normal distribution; the smaller Nu is, the tails of the distribution become heavier, allowing a better fit to data with more frequent extreme values. Due to all of the above, it was decided to use a Student’s t distribution to compare the means of the ACWR and acute load data. Subsequently, a generalized linear model (GLM) with a logistic link function was used to estimate the probability of injury given the belonging to different categories created according to exposure to load ranges the week prior to injury [39,40].

RStudio software (version 1.4.110) and the JAGS extension were used to estimate the model parameters through the Markov chain Monte Carlo (MCMC) method. Subsequently, the convergence of the chains was evaluated by means of graphical methods, effective sample size (ESS) estimation and Gelman–Rubin potential scale reduction factor estimation [41]. The posterior distributions of each relevant estimated parameter, its mean and 95% high-density interval were reported. The calculation of ACWR, ACWR with a synthetic denominator and the discretization of variables was performed. The ACWR was calculated as the ratio between the mean week load and the moving average of the last four weeks for each athlete.

In order to demonstrate that the ACWR does not contain explicit information about an athlete’s dynamic state, and in order to demonstrate that it is nothing more than a way of rescaling the acute load metric [42], a new ACWR was created for each observation, which was calculated as the ratio between the acute load and a random value generated from a normal distribution with a mean equal to the mean acute load of the whole team and deviation equal to the deviation of the acute load of the team. This new metric was named *ACWRr*. Its formula is shown below:$$ACWRr=\frac{Acute\;Load}{\gamma }$$
where *ACWRr* is the random acute-to-chronic workload ratio, acute load is the load of the week, and gamma is a random variable from a normal distribution with a mean of 1900 and a variance of 730. In this way, if the ACWR contains important information about the dynamic state of a subject, it should have a higher correlation with the probability of suffering an injury compared to its counterpart with a random denominator.

Since the *ACWR* and *ACWRr* are on different scales, it was decided to discretize both variables into 4 groups according to the quantiles of the data, which will facilitate the interpretation of the coefficients of the generalized linear model.

The acute workload was calculated as the sum of the daily workload for one week, and the cumulative workload was calculated as the sum of daily workloads for one, two, three and four weeks. Chronic workload was calculated as the 4-week rolling average acute workload

## 3. Results

The descriptive data of the variables used in the models can be seen in Table 1 and Table 2. Table 2 shows the distribution of data within each quantile created for the whole sample and for the subgroups according to the presence of injuries.

The results of the estimation of the parameters for the Student’s t distribution of AL and ACWR for athletes who did not suffer injuries and for those who did suffer injuries can be seen in Table 3 and Table 4. The estimated posterior distributions can be seen in Figure 1. In both cases for AL and ACWR, we can see that there is a significant difference between athletes who did and did not suffer injuries. In both cases, the distribution of the means shows great similarity, which is evident when calculating the effect size (ES) between groups, which turned out to be practically identical when calculated for AL (ES = 0.64, HDI 95% = (0.2, 1.09) and for ACWR (ES = 0.64, HDI 95% = (0.2, 1.08)). The mean difference of AL (Mean Diff = 449.23, HDI 95% = (146.41, 751.2) and ACWR (Mean Diff = 0.22, HDI 95% = (0.07, 0.36) between injured and non-injured athletes was also calculated and in both cases proved to be significant.

The results of the GLM model fit are shown in Table 5. In this case, the coefficient beta 0 represents the intercept and baseline for the sample. The rest of the coefficients represent the deviation with respect to the baseline, expressed in log odds, using as predictors the ACWR and ACWRr discretized in quantiles. Note that by being able to access the posterior distributions of the model coefficients, it is very easy to re-express the coefficients in terms of the conditional probability estimated by the model by replacing the coefficients in the link function used (i.e., logistic) [39]. The only coefficients that turned out to be significant (i.e., their 95% HDI does not contain zero) were the beta 4 coefficients. Figure 2 shows the distribution of the estimated coefficients for both models. Figure 3 shows the distribution of probability of injury given that one belongs to quantile 4 of ACWR (P (Inj | ACWR > 1.15) = 0.07, HDI 95% = (0.04, 0.11) and ACWRr (P (Inj | ACWR > 1.45) = 0.06, HDI 95% = (0.04, 0.1) [35].

## 4. Discussion

Our study aimed to investigate the relationship between injury risk, AL, ACWR, and a random ACWR created from replacing the ACWR denominator. In this regard, a general model was proposed to relate players’ ACWR and their potential injury risk using different loading units (i.e., running distance, accelerations, decelerations, or RPE) [30] and has been applied to teams in various sports [26,27,28,30]. Even in the literature, there has been an attempt to find a relationship between an optimal ACWR model, between 0.8 and 1.3, as a parameter for abrupt changes in load that could decrease the risk of injury [30].

In our case study, the main finding is that the increased ACWR and ACWRr also correlated with an increased probability of significant injury risk, although such an increase from baseline could be considered as “clinically irrelevant”; this may be because the model does not capture the complex interactions between ACWR and the variable to be predicted, or the predictors do not serve to explain the variable to be predicted. Consistent with our assertion, some studies tested the predictive ability of ACWR models and reported that, in isolation, they had poor or no ability to detect individuals who would suffer a sport-related injury [4,43], resulting in a high number of false-positive predictions [4,23].

Recently, it has been shown that the relationships between injuries and high ACWR are the result of artifacts in the statistical models used to evaluate the relationship between both variables, which amplify the estimated effect. These artifacts are usually the result of the use of ratios, which make the subsequent interpretation of the results of the models used more complex [42]. Other authors have already argued that the training adaptations developed by tissues cannot be adjusted to different types of averages. This approach does not accurately adjust the variations within the established time period and can thus lead to errors of interpretation, since they do not take into account the exact time at which the peak load may have occurred [44].

The occurrence of injury is a complex phenomenon [45] involving a large number of parameters (performance, workload, physiology, sleep, fatigue and recovery, psychology, lifestyle, etc.) and a large number of interactions between them that may even be more relevant in the outcome. The relationships between workload and injury are recursive and individualistic, with variables interacting in a non-linear fashion and with different levels of impact with different magnitudes of effects. For example, the physiological mechanisms that underlie the relationship between RPE, distance run, mode of exercise and injury are mediated by fitness level and fatigue, among others, and this relationship is different for the same athlete at another time. Consequently, small variations can generate large effects or even second-order effects that emerge from the interaction, as shown in the dynamics of complex systems [46]. In this sense, in our case, the cohort analysis shows that this model is uniquely unable to predict the occurrence of an injury more sensitively than a synthetically produced value.

A more clear etiology between sports injuries and training load is yet to be established [47]. Therefore, nowadays, the capacity of current metrics is limited. More specific approaches to each injury mechanism, looking for more concrete injury risk assessment [47].

The ACWR fails to capture an individual’s ability to adapt to a given stimulus, or the model fails to capture the complex interactions that determine an individual’s susceptibility to injury. The high similarity between the two pairs of distributions for the estimated means reinforces the claims that the ACWR is simply a way of rescaling the AL and does not provide more information. Using ACWR in practice may lead to inappropriate recommendations because its causal relationship to harm has not been proven, it is an imprecise measure (the numerator is not normalized by the denominator even when separated), and because of its causal role, which lacks background theory, an ambiguous indicator that is inconsistent and unilaterally related to injury risk [48]. The use of Bayesian estimates for our models clarifies errors made in the literature because their great flexibility facilitates the interpretation of the results obtained.

## 5. Conclusions

In conclusion, the ACWR ratio from internal load (IL) monitoring is no better than a random ACWR created from a random denominator to predict the probability of injury risk. This metric should not be used in isolation to analyze the causality between load and injury.

## Figures and Tables

**Figure 1 jcm-11-05945-f001:**
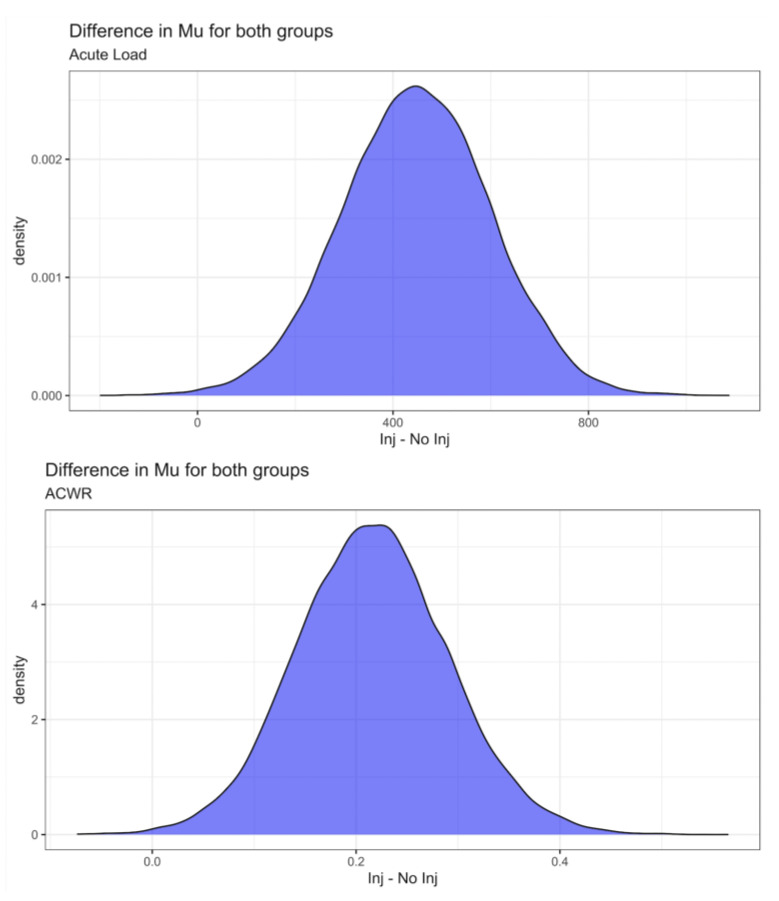
Posterior distribution of the mean difference between Injured and Not Injured for Acute Load (**Upper**) and ACWR (**Below**).

**Figure 2 jcm-11-05945-f002:**
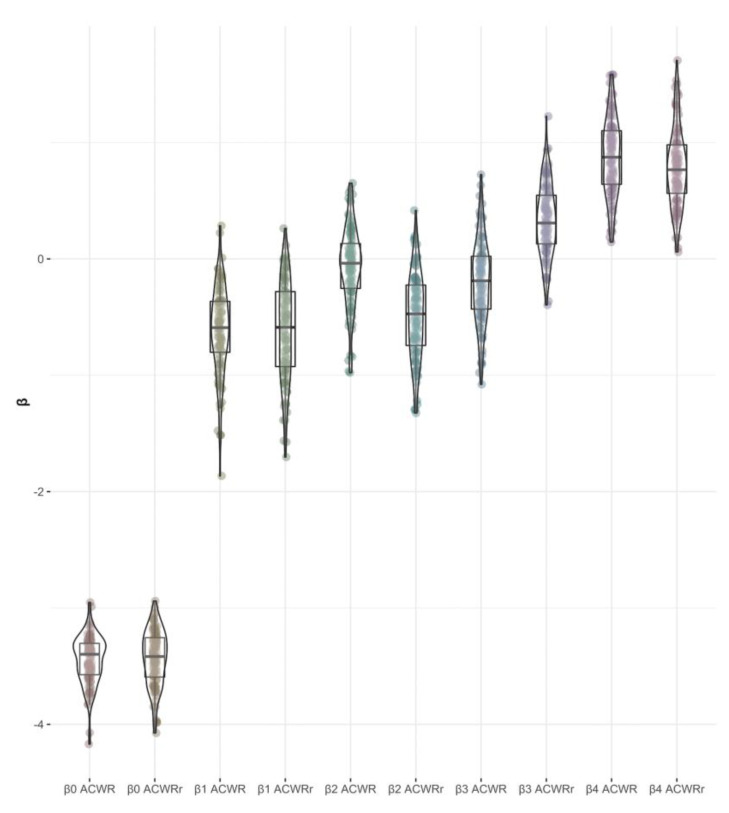
Violin plots representing posterior distribution for GLM coefficients.

**Figure 3 jcm-11-05945-f003:**
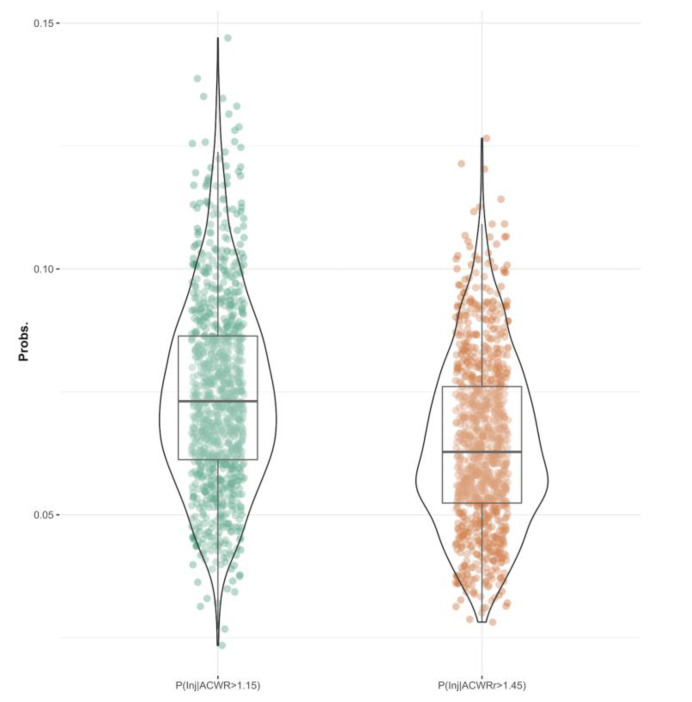
GLM prediction of injury for individuals in the fourth quantile for ACWR and ACWRr.

**Table 1 jcm-11-05945-t001:** Descriptive Statistics for Internal Load Variables.

	Overall, N = 815	Injured, N = 30	Not Injured, N = 785
ACWR	1 (0.76, 1.16)	1.2 (0.93, 1.35)	0.99 (0.76, 1.35)
ACWRr	0.98 (0.67, 1.46)	1.29 (1.02, 2.01)	0.97 (0.67, 2.01)
Acute Load	1800 (1400, 2308)	2138 (1820, 2880)	1780 (1400, 2880)
Cumulative Load (2 weeks)	3640 (3000, 4362)	3891 (3325, 4745)	3625 (2990, 4745)
Cumulative Load (3 weeks)	5331 (4513, 6271)	5458 (4582, 5899)	5327 (4511, 5899)
Cumulative Load (4 weeks)	7174 (5826, 8347)	7065 (5951, 8621)	7185 (5825, 8621)
Chronic Load (4 weeks)	1900 (1620, 2244)	1870 (1598, 2141)	1900 (1620, 2254)

Data shown as median (IQR); ACWR: acute chronic workload ratio; ACWRr: random acute chronic workload ratio.

**Table 2 jcm-11-05945-t002:** Data distribution in each quartile for the entire sample and for all subgroups.

Variables	Overall, N = 815	Injured, N = 30	Not Injured, N = 785
Acute Load Quantiles			
First	205 (25%)	3 (10%)	202 (26%)
Second	206 (25%)	5 (17%)	201 (26%)
Third	200 (25%)	8 (27%)	192 (24%)
Fourth	204 (25%)	14 (47%)	190 (24%)
ACWR Quantiles			
First	205 (25%)	3 (25%)	202 (26%)
Second	219 (27%)	6 (20%)	213 (27%)
Third	187 (23%)	5 (17%)	182 (23%)
Fourth	204 (25%)	16 (53%)	188 (24%)
ACWRr Quantiles			
First	204 (25%)	205 (25%)	201 (26%)
Second	204 (25%)	206 (25%)	200 (25%)
Third	203 (25%)	200 (25%)	194 (25%)
Fourth	204 (25%)	204 (25%)	190 (24%)

Data shown as count (%); ACWR: acute chronic workload ratio; ACWRr: random acute chronic workload ratio.

**Table 3 jcm-11-05945-t003:** Estimates of Student’s t parameters. Point estimates and HDI intervals for acute load.

	Mean	HDI 2.5%	HDI 97.5%	ESS
Injured				
Mu	2296.28	2000.21	2592.58	8196.6
Sigma	756.34	560.5	1023.57	3394.13
Not Injured				
Mu	1847.62	1792.49	1903.53	22,465.81
Sigma	654.33	590.28	720.85	14,753.83
Group Diff.				
Nu	12.46	5.23	38.46	4233.86
Effect Size	0.64	0.20	1.09	
Mean Diff	448.66	146.36	748.07	

Mu: location parameter; Sigma: scale parameter; Nu: degrees of freedom; HDI: high-density interval; ESS: effective sample size.

**Table 4 jcm-11-05945-t004:** Estimates of Student’s t parameters. Point estimates and HDI intervals for ACWR.

	Mean	HDI 2.5%	HDI 97.5%	ESS
Injured				
Mu	1.18	1.03	1.32	19,033
Sigma	0.38	0.28	0.51	6421
Not Injured				
Mu	0.96	0.94	0.98	17,854
Sigma	0.28	0.26	0.30	12,787
Group Diff.				
Nu	23.26	8.64	64.66	6155.5
Effect Size	0.64	0.2	1.08	
Mean Diff	0.21	0.07	0.36	

Mu: location parameter; Sigma: scale parameter; Nu: degrees of freedom; HDI: high-density interval; ESS: effective sample size.

**Table 5 jcm-11-05945-t005:** GLM logic link parameter estimates. Point estimates of HDI intervals for ACWR and random ACWR.

	Mean	HDI 2.5%	HDI 97.5%	ESS
ACWR				
β0	−3.45	−3.92	−3.04	11,557
β1	−0.62	−1.55	0.11	9272
β2	−0.13	−0.81	0.51	15,042
β3	−0.15	−0.87	0.53	15,573
β4	0.90	0.28	1.51	6059
P(Inj | ACWR > 1.15)	0.07	0.04	0.11	
Random ACWR				
β0	−3.43	−3.90	−3.03	10,439
β1	−0.62	−1.56	0.09	8005
β2	−0.41	−1.22	0.28	11,619
β3	0.30	−0.30	0.92	10,790
β4	0.73	0.12	1.34	5809
P(Inj | ACWRr > 1.45)	0.06	0.04	0.10	

ACWR: acute chronic workload ratio; ACWRr: random acute chronic workload ratio; HDI: high-density interval; ESS: effective sample size; (β0, β1, β2, β3, β4): Coefficients of the GLM; P (Inj | ACWR > 1.15): probability of suffering an injury given that ACWR is more than 1.15; P (Inj | ACWRr > 1.45): probability of suffering an injury given that ACWRr is more than 1.45.

## Data Availability

Data will be made available upon reasonable request to the corresponding author.

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
