# Peer review of "Is the Relationship between Acute and Chronic Workload a Valid Predictive Injury Tool? A Bayesian Analysis"

_jcm, 2022, doi:10.3390/jcm11195945_

Round 1

Reviewer 1 Report

GENERAL COMMENTS

In general, the topic of the article is very interesting and has a meaningful contribution to the field of research. But there are a few objectives and my personal opinions that should be taken into consideration by the author before considering it for publication.

The comments, broken down by manuscript sections, can be found below.

Generally,

the manuscript should be checked for grammar, spelling, and typos.

ABSTRACT

16, 17, 18 – consider rewording

HDI and AL – explain abbreviations when first mentioned

21 – synthetic – what do you mean by that?

25 – should, which metric?

The abstract is written poorly. It is hard to understand what you wanted to prove. Please consider rewording it and checking it for grammar and typos. In my opinion, the objective should be reworded. See my last comment below.

Why only an objective is emphasized? I think you need to delete “Objective”.

INTRODUCTION

line 29 – typo today’s //modern

34 – positively correlated/associated?

45 - >4mts/s2, replace with > 4 m/s^2

56 – typo: , [10]

Maybe you should define internal and external loading. And then describe the association between external and internal loading variables.

67 – intra session, inter-session means between the sessions

71 - perception, (session-RPE), - typo, delete comma

91-93 – please consider rewording, not clear enough, moreover what is a random workload?

94-96 – this was mentioned a few times in previous paragraphs. Please delete where unnecessary

107 – can you find another name for synthetic ACWR? Maybe adjusted/random ACWR? Optionally

 108-110 – this should not be a part of the introduction. Replace with your hypothesis – what did you expect and why did you do the study in the first place? You claim, that:  Associations have been found between high 103 ACWRs and increased injury risk in elite soccer[32], but no association has been found between increased ACWRs and increased injury risk in sports such as rugby. What is the contribution of your study to the field of research?

Also, see my last comment below.

The content of the introduction is suitable. The grammar must be checked and corrected.

METHODS

113 – consider rewording, it is not a replacement, it is a way to check the validity of the commonly used measure – ACWR!

122 - classification// definition replace with classification/definition

128 – type, body part, and injury mechanism

136, 140 - typo

Did you use all the parameters in the analysis? Type of injury, severity, mechanism… If not, you do not have to explain all of them.

Why did you decide to do a Bayesian analysis? Briefly explained in 144-146

167 – abbreviation of the equation is missing

169 – random value

160-172 – I do not understand exactly why did you use a random value from a normal distribution. From a practical perspective, this analysis is illogical. In general, when averaging the results of the whole group, the results should be the same as if the constant value was used.

New comment (after reading the whole article): please, emphasize that this was a way/an approach to check the validity of the old (classic ACWR) value.

RESULTS

177 – abbreviations are missing. How were cumulative and chronic loads calculated? Table 1,2 3,4,5. Check Figures also. Each table and figure should stand alone, complete and informative by itself.

196-205 – copy content to the methods section

DISCUSSION

233 - case//study: replace thought the text to only one “/”

229-232 – consider rewording

237, 240 – two spaces, typo – check through the text!

243-261 – I think that here is the answer to my question: “I do not understand exactly why did you use a random value from a normal distribution.” It was because you wanted to check the validity, specificity of the new measure (you named it synthetic ACWR). I suggest that you use terms validity, comparison and that you use the term “random ACWR” or something similar, instead of synthetic. The aim of the study is to check the validity of the ACWR value. And your aim is based on the fact, explained in 240-260. The aim of the study should be more clearly stated (last paragraph of the introduction, abstract and so on). The methods should also be more straightforwardly explained. For example: to check the validity of the commonly used measure “ACWR”, the measure was compared to the randomly adjusted ACWR using B. statistics. Write main outcomes in the first paragraph of the Discussion and answer to your hypothesis. The hypothesis should be mentioned in the last paragraph of the introduction. I think that you want to prove that ACWR is not a valid measure to distinguish between injured and uninjured athletes in the past year…

Author Response

RESPONSE – REVIEWER 1

In general, the topic of the article is very interesting and has a meaningful contribution to the field of research. But there are a few objectives and my personal opinions that should be taken into consideration by the author before considering it for publication.

The comments, broken down by manuscript sections, can be found below.

Authors’ response: We would like to thank the editor for taking time to provide last comments to improve the quality of the manuscript. In the following sections we answer point-by-point your final questions/suggestions.

Generally,

the manuscript should be checked for grammar, spelling, and typos.

 Authors’ response: Thank you for your comments. It has been modified accordingly.

ABSTRACT

16, 17, 18 – consider rewording

Authors’ response: Thank you for your comments. It has been modified accordingly.

HDI and AL – explain abbreviations when first mentioned

Authors’ response: Thank you for your comments. It has been modified accordingly.

21 – synthetic – what do you mean by that?

Authors’ response: Thank you for your comments. It has been modified accordingly (random ACWR)

25 – should, which metric?

Authors’ response: Thank you for your comments. It has been modified accordingly.

The abstract is written poorly. It is hard to understand what you wanted to prove. Please consider rewording it and checking it for grammar and typos. In my opinion, the objective should be reworded. See my last comment below.

Why only an objective is emphasized? I think you need to delete “Objective”.

Authors’ response: Thank you for your comments. It has been modified accordingly.

INTRODUCTION

line 29 – typo today’s //modern

Authors’ response: Thank you for your comments. It has been modified accordingly.

34 – positively correlated/associated?

Authors’ response: Thank you for your comments. It has been modified accordingly.

45 - >4mts/s2, replace with > 4 m/s^2

Authors’ response: Thank you for your comments. It has been modified accordingly.

56 – typo: , [10]

Authors’ response: Thank you for your comments. It has been modified accordingly.

Maybe you should define internal and external loading. And then describe the association between external and internal loading variables.

Authors’ response: Thank you for your comments. This concept has been expanded.

67 – intra session, inter-session means between the sessions

Authors’ response: Thank you for your comments. It has been modified accordingly.

71 - perception, (session-RPE), - typo, delete comma

Authors’ response: Thank you for your comments. It has been modified accordingly.

91-93 – please consider rewording, not clear enough, moreover what is a random workload?

94-96 – this was mentioned a few times in previous paragraphs. Please delete where unnecessary

Authors’ response: Thank you for your comments. It has been modified accordingly.

107 – can you find another name for synthetic ACWR? Maybe adjusted/random ACWR? Optionally

Authors’ response: Thank you for your comments. It has been modified accordingly (random ACWR)

 108-110 – this should not be a part of the introduction. Replace with your hypothesis – what did you expect and why did you do the study in the first place? You claim, that:  Associations have been found between high 103 ACWRs and increased injury risk in elite soccer [32], but no association has been found between increased ACWRs and increased injury risk in sports such as rugby. What is the contribution of your study to the field of research?

Also, see my last comment below.

The content of the introduction is suitable. The grammar must be checked and corrected.

METHODS

113 – consider rewording, it is not a replacement, it is a way to check the validity of the commonly used measure – ACWR!

122 - classification// definition replace with classification/definition

128 – type, body part, and injury mechanism

136, 140 - typo

Did you use all the parameters in the analysis? Type of injury, severity, mechanism… If not, you do not have to explain all of them.

Why did you decide to do a Bayesian analysis?

 Briefly explained in 144-146

167 – abbreviation of the equation is missing

Authors’ response: Thank you for your comments, abbreviation was added below.

169 – random value

160-172 – I do not understand exactly why did you use a random value from a normal distribution. From a practical perspective, this analysis is illogical. In general, when averaging the results of the whole group, the results should be the same as if the constant value was used.

New comment (after reading the whole article): please, emphasize that this was a way/an approach to check the validity of the old (classic ACWR) value.

Authors’ response: Thank you for your comments. A brief description was added 201-203

RESULTS

177 –. How were cumulative and chronic loads calculated?

Authors’ response: Thank you for your comments. Description for the calculation of cumulative and chronic workload was added.

Abbreviations are missing  Table 1,2 3,4,5. Check Figures also. Each table and figure should stand alone, complete and informative by itself.

Authors’ response: Thank you for your comments. Description for each abbreviation was added

196-205 – copy content to the methods section

 Authors’ response: Thank you for your comments. The paragraph was reformulated and added in methods

DISCUSSION

233 - case//study: replace thought the text to only one “/”

Authors’ response: Thank you for your comments. It has been modified accordingly.

229-232 – consider rewording

Authors’ response: Thank you for your comments. It has been modified accordingly.

237, 240 – two spaces, typo – check through the text!

Authors’ response: Thank you for your comments. It has been modified accordingly.

243-261 – I think that here is the answer to my question: “I do not understand exactly why did you use a random value from a normal distribution.”

 It was because you wanted to check the validity, specificity of the new measure (you named it synthetic ACWR). I suggest that you use terms validity, and comparison and that you use the term “random ACWR” or something similar, instead of synthetic. The aim of the study is to check the validity of the ACWR value. And your aim is based on the fact, explained in 240-260.

The aim of the study should be more clearly stated (last paragraph of the introduction, abstract and so on). The methods should also be more straightforwardly explained. For example: to check the validity of the commonly used measure “ACWR”, the measure was compared to the randomly adjusted ACWR using B. statistics. Write main outcomes in the first paragraph of the Discussion and answer to your hypothesis. The hypothesis should be mentioned in the last paragraph of the introduction.

 Authors’ response: Thank you for your comments. We improved the last part of the introduction with the aim of being more clear.

I think that you want to prove that ACWR is not a valid measure to distinguish between injured and uninjured athletes in the past year…

Reviewer 2 Report

In general, the manuscript is well written and the has a solid argumentation and rational. Why I see that the paper adds something to the scientific community in analyzing ACWR data with an Bayesian approach as all the literature / simulation studies on this topic use as frequentist idea, the authors need to highlight this more in the introduction.  As the introduction stands, it is hard to see as a “normal” reader, what the paper adds.

To do so, I would shorten the more general sections on RPE line 66- 81 and give more space to the discussion on ACWR. The authors should highlight that plenty of authors did find a relationship with ACWR and injuries using small sample size and simplistic statistics, as highlighted by Impellizzeri.

They might also add the ideas of this group to solve this problem:

Jeffries, A.C., Marcora, S.M., Coutts, A.J. et al. Development of a Revised Conceptual Framework of Physical Training for Use in Research and Practice. Sports Med 52, 709–724 (2022). https://doi.org/10.1007/s40279-021-01551-5

Impellizzeri FM, Menaspà P, Coutts AJ, Kalkhoven J, Menaspà MJ. Training Load and Its Role in Injury Prevention, Part I: Back to the Future. J Athl Train. 2020 Sep 1;55(9):885-892. doi: 10.4085/1062-6050-500-19. PMID: 32991701; PMCID: PMC7534945.

The only methodological issue I have with the paper is the assumption of T-Student distribution. The load data, the authors are working with, are normally skewed. Therefore, one would rather use a distribution that accounts for that. See an example below, using a similar approach as the authors:

Hamid Zareifard, Majid Jafari Khaledi, A heterogeneous Bayesian regression model for skewed spatial data, Spatial Statistics, Volume 46, 2021, https://doi.org/10.1016/j.spasta.2021.100545

Minor:

You have this paper twice in your references:

Impellizzeri, F., et al., Acute to random chronic workload ratio is‘as’ associated with injury as acute to actual chronic workload ratio: time to dismiss ACWR and its components. SportRXiv, 2020.

With these changes in the introduction and having it reflected in the discussion, I think this is a solid paper.

Author Response

RESPONSE – REVIEWER 2

In general, the manuscript is well written and the has a solid argumentation and rational. Why I see that the paper adds something to the scientific community in analyzing ACWR data with an Bayesian approach as all the literature / simulation studies on this topic use as frequentist idea, the authors need to highlight this more in the introduction.  As the introduction stands, it is hard to see as a “normal” reader, what the paper adds.

Authors’ response: We would like to thank the editor and reviewer for taking time to provide last comments to improve the quality of the manuscript. In the following sections we answer point-by-point your final questions/suggestions.

To do so, I would shorten the more general sections on RPE line 66- 81 and give more space to the discussion on ACWR. The authors should highlight that plenty of authors did find a relationship with ACWR and injuries using small sample size and simplistic statistics, as highlighted by Impellizzeri.

Authors’ response: Thank you for your comments. It has been modified accordingly throughout the different sections of the text

They might also add the ideas of this group to solve this problem:

Jeffries, A.C., Marcora, S.M., Coutts, A.J. et al. Development of a Revised Conceptual Framework of Physical Training for Use in Research and Practice. Sports Med 52, 709–724 (2022). https://doi.org/10.1007/s40279-021-01551-5

Impellizzeri FM, Menaspà P, Coutts AJ, Kalkhoven J, Menaspà MJ. Training Load and Its Role in Injury Prevention, Part I: Back to the Future. J Athl Train. 2020 Sep 1;55(9):885-892. doi: 10.4085/1062-6050-500-19. PMID: 32991701; PMCID: PMC7534945.

The only methodological issue I have with the paper is the assumption of T-Student distribution. The load data, the authors are working with, are normally skewed. Therefore, one would rather use a distribution that accounts for that. See an example below, using a similar approach as the authors:

Authors’ response: Thank you very much for this point, it is true that the reason for using a T-Student distribution was not clearly explained. A paragraph was added narrating how and why this election was decided.

Hamid Zareifard, Majid Jafari Khaledi, A heterogeneous Bayesian regression model for skewed spatial data, Spatial Statistics, Volume 46, 2021, https://doi.org/10.1016/j.spasta.2021.100545

Minor:

You have this paper twice in your references:

Impellizzeri, F., et al., Acute to random chronic workload ratio is‘as’ associated with injury as acute to actual chronic workload ratio: time to dismiss ACWR and its components. SportRXiv, 2020.

Authors’ response: Thank you for your comments. It has been modified accordingly.

With these changes in the introduction and having it reflected in the discussion, I think this is a solid paper.

Reviewer 3 Report

Thanks for give me the possibility to review the current manuscript. In general, the topic of the article is very interesting, and the manuscript shows remarkable data and findings. Just minor things must be taken into consideration by the authors. I recommend that the manuscript must be reviewed by a native speaker. 

Authors wrote “high intensity accelerations and decelerations, around 25 accelerations (>4mts/s2) and 45 very high intensity decelerations (>-4mts/s2)”. Why accelerations > 4 m/s/s are not very high intensity accelerations? Please change >4mts/s2.

I suggest explaining the readers the difference between internal and external loads.

Line 71 introduction, please rephrase.

Please define in the introduction a clear hypothesis and objectives.

Please include in methods more info about the injuries suffered by players.

Use the abbreviations during the manuscript. Several time abbreviations are missing.

Check all the paragraphs during the manuscript (2 spaces?). Be consistent.

Please include the main findings of the study at the beginning of the discussion after the aim.

Check the references. A paper is twice.

Author Response

RESPONSE – REVIEWER 3

Thanks for give me the possibility to review the current manuscript. In general, the topic of the article is very interesting, and the manuscript shows remarkable data and findings. Just minor things must be taken into consideration by the authors. I recommend that the manuscript must be reviewed by a native speaker. 

Authors’ response: We would like to thank the editor and reviewer for taking time to provide last comments to improve the quality of the manuscript. In the following sections we answer point-by-point your final questions/suggestions.

Authors wrote “high intensity accelerations and decelerations, around 25 accelerations (>4mts/s2) and 45 very high intensity decelerations (>-4mts/s2)”. Why accelerations > 4 m/s/s are not very high intensity accelerations? Please change >4mts/s2.

Authors’ response: Thank you for your comments. It has been modified accordingly.

I suggest explaining the readers the difference between internal and external loads.

Line 71 introduction, please rephrase.

Authors’ response: Thank you for your comments. We have explained both concepts and its differences.

Please define in the introduction a clear hypothesis and objectives.

Authors’ response: Thank you for your comments. These concepts have been expanded.

Please include in methods more info about the injuries suffered by players.

Authors’ response: Thank you for your comments. This concept have been expanded.

Use the abbreviations during the manuscript. Several time abbreviations are missing.

Authors’ response: Thanks for your suggestion. It has been modified accordingly

Check all the paragraphs during the manuscript (2 spaces?). Be consistent.

Authors’ response: Thanks for your suggestion. It has been modified accordingly

Please include the main findings of the study at the beginning of the discussion after the aim.

Authors’ response: Thank you for your comments. We have included the required concept

Check the references. A paper is twice

Authors’ response: Thanks for your suggestion. It has been modified accordingly

Round 2

Reviewer 1 Report

The manuscript was improved and now it meets the criteria for publication. Nevertheless, I recommend an additional English language check.

Best regards